# Lifestyle Medicine for the Prevention and Treatment of Pancreatitis and Pancreatic Cancer

**DOI:** 10.3390/diagnostics14060614

**Published:** 2024-03-14

**Authors:** Sruthi Kapliyil Subramanian, Bhaumik Brahmbhatt, Jennifer M. Bailey-Lundberg, Nirav C. Thosani, Pritesh Mutha

**Affiliations:** 1Center for Interventional Gastroenterology at UTHealth (iGUT), Section of Endoluminal Surgery and Interventional Gastroenterology, Division of Elective General Surgery, Department of Surgery, McGovern Medical School at UTHealth, Houston, TX 77030, USA; sruthi.kapliyil.subramanian@uth.tmc.edu (S.K.S.); pritesh.r.mutha@uth.tmc.edu (P.M.); 2Mayo Clinic, Division of Gastroenterology and Hepatology, Jacksonville, FL 32224, USA; brahmbhatt.bhaumik@mayo.edu; 3Department of Anesthesiology, Critical Care and Pain Medicine, McGovern Medical School at UTHealth, Houston, TX 77030, USA; jennifer.m.bailey@uth.tmc.edu

**Keywords:** pancreatitis, pancreatic cancer, whole food plant based diet, lifestyle management

## Abstract

The incidence of pancreatitis and pancreatic cancer is on the upswing in the USA. These conditions often lead to higher healthcare costs due to the complex nature of diagnosis and the need for specialized medical interventions, surgical procedures, and prolonged medical management. The economic ramification encompasses direct healthcare expenses and indirect costs related to productivity losses, disability, and potential long-term care requirements. Increasing evidence underscores the importance of a healthy lifestyle in preventing and managing these conditions. Lifestyle medicine employs evidence-based interventions to promote health through six key pillars: embracing a whole-food, plant-predominant dietary pattern; regular physical activity; ensuring restorative sleep; managing stress effectively; removing harmful substances; and fostering positive social connections. This review provides a comprehensive overview of lifestyle interventions for managing and preventing the development of pancreatitis and pancreatic cancer.

## 1. Introduction

Acute pancreatitis is an inflammatory disease characterized by abdominal pain and elevated levels of pancreatic enzymes in the blood. The global incidence of acute pancreatitis is 34 affected individuals per 100,000 people each year and has an associated mortality rate of 1–5%. The incidence of acute pancreatitis has been increasing worldwide and directly causes a healthcare system-associated cost of USD 9.3 billion annually [1,2]. In the United States, seventy-five percent of the cases of acute pancreatitis are due to gallstones and chronic alcohol exposure, indicating areas for therapeutic intervention and lifestyle modifications to reduce disease burden and healthcare costs.

If not appropriately managed, acute pancreatitis can lead to recurrent acute pancreatitis, which can result in the development of chronic pancreatitis. Chronic pancreatitis is a severe inflammatory disease derived from permanent structural damage and impairment of both endocrine and exocrine functions. The global incidence of chronic pancreatitis ranges from 5 to 12 affected individuals per 100,000 people each year, with complications including pseudocyst formation and obstruction of the duodenum and common bile duct [3]. Chronic pancreatitis is the most substantial identified risk factor for pancreatic cancer.

Pancreatic adenocarcinoma (PDAC) carries the lowest survival rate of all organ cancers and is the third-leading contributor to cancer mortality in the United States. Following diagnosis, survival typically ranges from 4 to 6 months [4]. Because of the low survival associated with pancreatic cancer, incidence and mortality rates nearly mirror each other. The reasons for this low survival rate include an extremely aggressive disease course, lack of diagnostic tools for early detection, and limited understanding of its etiology.

The incidence of pancreatic cancer has increased over recent decades and will become the second-leading cause of cancer-related deaths by the year 2030 [5]. Some proposed reasons for the observed increase in incidence rates include increased rates of tobacco smoking, obesity, diabetes mellitus, physical inactivity, and consumption of high-calorie and high-fat diets in certain countries, as well as improvements in the clinical recognition and diagnosis of pancreatic cancer, and the increasing life expectancy of the global population [6].

Epidemiological studies show a strong association between PDAC and inflammatory stimuli. Studies of dietary patterns and cancer outcomes suggest that diet and lifestyle changes might influence an individual’s risk of PDAC by modulating metabolic fluctuations and inflammation [7]. With the difficulty of diagnosis and low survival rate of pancreatic cancer, prevention with lifestyle medicine may be the best way to reduce the incidence of pancreatic cancer.

Lifestyle medicine is a medical specialty that focuses on an evidence-based therapeutic use of lifestyle changes, which consist of six pillars—a whole-food, plant-predominant eating pattern; physical activity; restorative sleep; stress management; the avoidance of risky substances; and positive social connections—emphasizing the prevention, treatment, and reversal of chronic diseases (Figure 1).

Lifestyle medicine is critical in understanding, preventing, and treating pancreatitis and pancreatic cancer. By focusing on the six pillars of lifestyle medicine, individuals can adopt healthy behaviors that promote their overall well-being and potentially reduce the risk or impact of these conditions. In this review, we will provide details within each of the six pillars and discuss how they affect the prevention and treatment of pancreatitis and pancreatic cancer (Table 1).

### 1.1. Whole-Food Plant-Predominant Diet

Recent extensive meta-analysis studies have emphasized the health benefits of a plant-based whole-food diet. Concerning cancers of the digestive system, one notable meta-analysis revealed that plant-based diets significantly reduce the risk of developing digestive neoplasms in the pancreas, colon, and rectum [8]. Interestingly, this large study found no discernible difference in this correlation between vegan and other plant-based diets. When focusing specifically on pancreatitis and pancreatic cancer risks, adopting a whole-food-based diet emphasizing fruits, vegetables, beans, and whole grains demonstrated reduced incidence of both conditions [9]. Another study on a U.S. population-based cohort of 101,748 adults revealed that diets that scored highly on a plant-based scale were associated with the lowest risk of developing pancreatic cancer [10]. Based on multiple comprehensive meta-analyses, the Mediterranean diet, abundant in fresh fruits and vegetables, has been identified as an optimal anti-cancer diet [11]. Additionally, Mediterranean and other plant-based diets are known to possess anti-inflammatory properties and may also inhibit the growth rate of cancer cells by inducing apoptosis [12].

**(A)** 
**Phytates:**


Phytate is a hexakisphosphate or inositol polyphosphate, serving as the primary reservoir of phosphorus in various plant tissues, particularly bran and seeds. It is alternatively known as inositol hexakisphosphate. Phytates found in whole grains, beans, nuts, and seeds assimilate into the bloodstream and are captured by tumor cells. Phytates exhibit their effects on tumor cells through a combination of antioxidative, anti-inflammatory, and immune-stimulating activities and by enhancing the function of natural killer cells [11]. Phytates have demonstrated inhibitory properties against a broad spectrum of human cancer cells tested so far, encompassing colon, breast, cervix, prostate, liver, pancreas, and skin cancers, while leaving normal cells unaffected. This characteristic distinguishes them as an outstanding anti-cancer agent, capable of discerning between tumor cells and healthy tissue [13,14]. Besides their recognized anti-cancer and immunomodulatory attributes, numerous phytonutrients present in plants can impede the development of new blood vessels that supply nutrients to tumors and disrupt existing tumor blood flow, thereby starving cancerous tumors. Additionally, phytates can occasionally induce cancer cells to revert to their normal state, ceasing their malignant behavior [15].

**(B)** 
**Methionine restriction:**


Methionine is an essential amino acid that plays a vital role in the growth and metabolism of cancer cells. It is primarily found in animal-derived products such as lean meat and eggs, while plant-based sources contain lower levels. Cancer cells necessitate a constant supply of methionine since they cannot recycle it from homocysteine-like healthy cells. Research has indicated that restricting methionine intake can impede cancer cell growth and potentially enhance the effectiveness of chemotherapeutic agents. Adopting a plant-based diet can achieve this goal [16]. Furthermore, studies have demonstrated that dietary methionine restriction in animals or cell culture media yields metabolic benefits, including decreased adiposity [17,18], increased insulin sensitivity [18,19], reduced inflammation [20] and oxidative stress [21], and extended lifespan [22].

In a study utilizing a patient-derived orthotopic xenograft nude mouse model of pancreatic cancer, a combination of recombinant methioninase to induce methionine restriction and the chemotherapeutic agent gemcitabine exhibited greater efficacy in treating pancreatic cancer compared to using gemcitabine alone. This combination also helped in overcoming gemcitabine resistance [23].

**(C)** 
**Plant protein vs. animal protein:**


Adopting a plant-based diet has been demonstrated to lower IGF-1 (insulin-like growth factor-1) levels while increasing the presence of the IGF-1 binding protein. IGF-1 is a growth hormone associated with cancer promotion, contributing to the genesis and progression of malignant tumors. When its presence is increased, the IGF-1 binding protein exhibits a protective mechanism by binding to excessive IGF-1, preventing excessive growth [24,25]. Studies have indicated that pancreatic cancer tissues often exhibit elevated levels of IGF-1 and IGF-1 receptors [26]. Immunohistochemical investigations have shown that insulin-like growth factor-I receptor (IGF1R) overexpression is linked to poor prognosis in patients with pancreatic ductal adenocarcinoma [27].

Animal proteins can stimulate the production of IGF-1. Epidemiological studies have revealed that individuals aged 50–65 years who reported high animal protein intake experienced a 75% increase in overall mortality and a four-fold increase in the risk of death from cancer during the subsequent 18 years. These associations were either eliminated or mitigated when the proteins consumed were of plant origin [28].

**(D)** 
**Fiber:**


Dietary fiber is a carbohydrate in plant foods such as whole grains, vegetables, fruit, and legumes, which have been dominant in human diets for millions of years. Dietary fiber, especially from whole foods, is crucial in promoting a healthy gut microbiota. Human digestive enzymes do not digest dietary fiber and thus it reaches the colon, where it interacts with the gut microbiota. Fiber is then fermented and metabolized by the gut microbiota, breaking it into compounds like short-chain fatty acids (SCFAs), including butyrate, acetate, and propionate. SCFAs help maintain the integrity of the gut lining, reduce inflammation, and regulate the immune system. Butyrate, mainly, is known for its anti-inflammatory properties and role in promoting gut health [29]. Fiber has the potential to positively impact the composition of the intestinal microbiota, which plays a role in regulating immune and inflammatory responses, thus further lowering the risk of cancer [30,31]. It is well documented that dietary habits, particularly those rich in fiber, can influence the structure and diversity of the gut microbiome. A diet high in fiber can lead to greater microbial diversity and a more balanced microbial composition [32].

A fiber-rich diet is associated with improved metabolic health, including better blood sugar control and reduced risk of obesity. Changes in the gut microbiota mediate these effects. Emerging evidence suggests that alterations in the human gut microbiome can play a role in the development and progression of a wide range of complex and chronic conditions, including obesity, diabetes, various types of cancers, and cardiovascular disease [33,34,35,36]. A case–control study conducted in Italy involving 326 pancreatic cancer patients found that both soluble and insoluble fiber derived from fruits showed an inverse association with pancreatic cancer [37]. Another study focusing on the Mediterranean diet demonstrated a protective effect of whole grains against various cancers, including pancreatic cancer [38]. Furthermore, a prospective study involving 75,680 women (Nurses’ Health Study) revealed that individuals who consumed 28 g of nuts twice or more every week had a significantly reduced risk of developing pancreatic cancer [39]. To promote a healthier gut microbiota, consuming a variety of fiber-rich whole foods such as fruits, vegetables, whole grains, legumes, nuts, and seeds is essential. Highly processed foods often lack fiber and may not provide the same benefits for gut health.

**(E)** 
**Curcumin:**


Curcumin is a yellow pigment predominantly found in turmeric and belongs to the polyphenol group. It possesses anti-inflammatory properties and can stimulate the body’s antioxidant production. Through preclinical in vitro and in vivo studies, curcumin has been found to exhibit various pharmacological effects, including antioxidant, anti-inflammatory, and anti-cancer activities, in different types of cancer, including pancreatic cancer. Multiple signaling pathways modulate these effects [40,41,42].

One study conducted at the MD Anderson Cancer Center involved administering high doses of curcumin to patients with advanced pancreatic cancer. Out of the twenty-one patients studied, two demonstrated positive responses to the treatment. One patient experienced a 73% reduction in tumor size, but this response was short-lived, as a resistant clone emerged. Another patient exhibited a stable course of disease for almost 18 months and their condition gradually improved over 1.5 years [43]. Notably, disease markers increased when physicians stopped the curcumin treatment for three weeks. Curcumin down-regulated the expression of NF-κB, cyclooxygenase-2, and phosphorylated signal transducer and activator of transcription 3 in peripheral blood mononuclear cells from patients, many of whom had initial levels significantly higher than those found in healthy volunteers. No curcumin-related toxic effects were observed, even with doses of up to 8 g daily.

**(F)** 
**Meat:**


There is a notable link between the consumption of well-done meat and the risk of pancreatic cancer. When cooked at high temperatures, meat can generate carcinogenic compounds known as heterocyclic amines (HCAs) and polycyclic aromatic hydrocarbons (PAHs). Meats cooked using high-temperature methods like barbecuing, grilling, and well-done preparations have increased levels of HCAs and PAHs. Consuming higher amounts of dietary mutagens, including HCAs and PAHs, has been associated with a two-fold increased risk of pancreatic cancer among individuals without a family history of cancer [44].

**(G)** 
**Animal fat:**


Dietary fat of animal origin increases the risk of pancreatic cancer risk [45]. When fats and fatty acids from ingested food reach the duodenum, the release of cholecystokinin stimulates the secretion of pancreatic enzymes and leads to pancreatic hypertrophy and hyperplasia. These processes can potentially increase the susceptibility of the pancreas to other carcinogens [46].

Furthermore, various observational studies and randomized controlled trials have demonstrated an association between saturated fat consumption and insulin resistance [47]. Diabetes and insulin resistance have been found to be related to an increased risk of pancreatic cancer in both epidemiological and animal studies [48].

**(H)** 
**Poultry:**


Studies have also indicated that poultry consumption is associated with an increased risk of pancreatic cancer. The relative increase in risk per 50 g of consumption per day is 1.72, with a 95% confidence interval of 1.04–2.84 [49]. Concerns have been raised about the potential transmission of avian viruses, such as Marek’s disease virus (MDV), to humans while handling fresh or frozen chicken. MDV is a herpesvirus that causes T-cell lymphoma in chickens. To assess the potential cancer-causing effects of these poultry viruses in humans, studies that evaluate populations with significant exposure should be conducted, similar to studying the effects of asbestos exposure [50]. Research on poultry workers has shown that individuals exposed to high levels of poultry oncogenic viruses are at a higher risk of dying from various cancers [51]. One case–control study examining workers in a poultry farm found that those involved in chicken slaughter had approximately nine times the odds of developing pancreatic cancer [52]. While the intensity of exposure to these viruses in the general population may not be as high as that experienced by poultry workers, the general population is still widely exposed to these viruses through contact with live poultry, blood or secretions from poultry, and raw meat. Additionally, exposure can occur through ingesting raw or inadequately cooked poultry meat or eggs.

**(I)** 
**Iron:**


A higher heme iron intake has a positive association with cancer risk. Iron is classified into either heme iron (organic) or non-heme iron (inorganic). Red meat contains heme iron. Heme iron has a higher absorption rate (between 15% and 40%) than non-heme iron, which is present in animal and plant sources (mainly in whole grains, legumes, and certain vegetables) and has a broader absorption range. When heme iron is in excess, it can exist as free iron. Free iron can generate reactive oxygen species (ROS), such as hydroxyl radicals, which can induce lipid peroxidation and cause oxidative damage to DNA. Additionally, heme iron, predominantly found in red meat, may contribute to carcinogenesis by acting as a nitrosating agent and forming N-nitroso compounds, which are known carcinogens [53]. Numerous large prospective cohort studies have demonstrated an association between the intake of red meat, the primary source of heme iron, and pancreatic cancer [54].

In conclusion, adopting a plant-based diet rich in whole foods and low in animal products, mainly red and processed meat, can significantly reduce the risk of pancreatic cancer and promote overall health. Additionally, incorporating specific dietary components like phytates, methionine restriction, dietary fiber, curcumin, and avoiding well-done meat and heme iron can further contribute to a lower risk of this type of cancer.

### 1.2. Physical Activity

Sarcopenia is a common complication of chronic pancreatitis and increases the likelihood of developing premature frailty in this population [55,56]. This condition refers to the degenerative loss of skeletal muscle mass, quality, and strength. Engaging in regular physical activity and maintaining functional fitness through aerobic and resistance exercises have been shown to mitigate the risks and effects of sarcopenia [57]. One study demonstrated that a 12-week yoga-based exercise intervention improved the quality of life and reduced stress indicators in patients with chronic pancreatitis compared to those who did not receive any intervention [58]. Physical activity reduces abdominal fat depots, leading to metabolic improvements in glucose tolerance and insulin sensitivity, thus lowering the risk of pancreatic cancer [59]. Excessive abdominal fat is associated with higher levels of C-peptide and circulating insulin, which are markers of hyperinsulinemia. These markers are linked to a two-fold increased risk of pancreatic cancer [60]. In the hyperinsulinemic state, exocrine cells of the pancreas are exposed to high insulin concentrations due to their blood supply passing through the islet cell region [61]. Such elevated insulin levels can activate the IGF-1 receptor, potentially promoting aberrant cell proliferation and the oncogenic transformation of pancreatic ductal cells. Hyperinsulinemia may also decrease the levels of IGF-1 binding protein, thereby increasing the bioavailability of IGF-1, which further supports cell proliferation [62]. One sizeable case–control study indicated that women in the highest quartile of BMI (≥27.2 kg/m^2^) had a 50% greater risk of pancreatic cancer, while men in the highest quartile (>34.4 kg/m^2^) had a 60% greater risk, compared to those in the lowest quartiles [63].

### 1.3. Stress Management

Stress triggers physiological responses that affect various systems in the body, particularly the sympathetic nervous system, the hypothalamic–pituitary–adrenal (HPA) axis, and the immune system [64]. These responses involve the release of neurotransmitters such as noradrenaline, adrenaline, and cortisol, which have been found to promote the growth of pancreatic cancer (PaCa) in mouse xenograft models [65].

A cohort study conducted in Sweden revealed that severe emotional stress, such as the loss of a parent, was associated with an increased risk of early-onset pancreatic cancer (occurring before the age of 40), regardless of the individual’s age at the time of the loss. Pancreatic cancer demonstrated the strongest association with parental death among all the cancer types [66]. Similar results were observed for the incidence of pancreatic cancer following the loss of a child [67].

Research has shown that non-selective β-blockers, commonly used to treat chronic stress and depression, can reduce pancreatic cancer progression in patients without metastasis [68,69]. This suggests a potential therapeutic benefit of these medications in pancreatic cancer management. The pancreas receives innervation from both sympathetic and parasympathetic nerves, with higher nerve density observed in tumor tissue. Stress induces the overexpression of β-adrenergic signaling, leading to increased levels of neurotrophins such as nerve growth factor (NGF) and brain-derived neurotrophic factor (BDNF). This contributes to the interaction between nerves and tumors through axogenesis, which is associated with higher tumor aggressiveness and a poorer prognosis in pancreatic cancer [70,71]. In a mouse model, stress has been shown to promote pancreatic cancer progression by compromising immune system activity. This includes reduced cytokine production, interferon-gamma (IFN-γ) levels, and interleukins, and a decrease in the population of T-lymphocytes (CD4 cells). Moreover, chronically stressed mice exhibited elevated levels of transforming growth factor beta (TGF-β) and vascular endothelial growth factor (VEGF), leading to increased PaCa growth and metastasis [72].

Mindfulness meditation has become a popular way to help people manage stress and improve their overall well-being. This is a technique that involves training your attention to the present moment and accepting your feelings and sensations without judgment [73]. This mainly involves breathing exercises, yoga, and guided lessons to help people become aware of their bodily sensations, thoughts, and feelings. It is believed that mindfulness can decrease the body’s response to stress, helping with overall physical and mental health outcomes. It changes the brain structures and activity in the regions associated with attention and emotion regulation, which helps people reduce negative thoughts and unhelpful emotional reactions during stress [74]. It has also been shown to improve the quality of life in several cancers [75]. A large meta-analysis examining 209 studies with a combined total of 12,145 participants of diverse ages, genders, and clinical profiles showed strong and clinically significant effects of treating anxiety, depression, and stress with mindfulness-based therapy [76]. Mindfulness can also be used in the treatment of cachexia and malnutrition in patients with cancer [77]. Mindfulness meditation can be a valuable complementary practice for individuals diagnosed with pancreatic cancer, as it can help improve their overall well-being and quality of life.

Mindfulness meditation causes significant improvements in chronic pain in both experimental and clinical settings [78]. Studies have shown that with mindfulness meditation, the systematic cultivation of a flexible attentional capacity for detached observation of proprioception can enhance whatever a patient’s existing coping strategies are and reduce their level of distress [79]. This can be a helpful tool for managing the chronic pain associated with chronic pancreatitis and help wean these patients off opioids. While it may not be a cure, mindfulness meditation can provide relief and improve these patients’ overall quality of life.

### 1.4. Restorative Sleep to Maintain Circadian Rhythm

Sleep disorders are prevalent in the general population, with insomnia (6–20%) [80], hypersomnia (0.5% to 1.6%) [81], and REM sleep behavior disorder (3–10%) [82] being the most common. Sleep is crucial in maintaining optimal immune, cellular, metabolic, and endocrine functioning. Neurophysiological system dysfunctions can promote cancer; therefore, it is plausible that disruptions in the sleep system may directly contribute to the risk of developing cancer.

Recent research suggests that exposure to light at night (LAN), which disrupts the circadian rhythm, may be a risk factor for cancer. LAN has also been associated with obesity and diabetes, which are known risk factors for pancreatic ductal adenocarcinoma. Night-shift work has been classified as a probable carcinogen for humans by the International Agency for Research on Cancer [83] and has been linked to more than a two-fold increase in the risk of pancreatic cancer in men [84]. Circadian disruption also negatively impacts various biological pathways involved in tumorigenesis, including immune function, hormone release, cell proliferation, and the cellular response to DNA damage [85,86].

One epidemiological study comparing individuals living in areas with the lowest LAN exposure to those with the highest LAN exposure showed a 27% increased risk of pancreatic ductal adenocarcinoma (PDAC) in those living with the highest LAN exposure. The trouble was similar for both men and women, with a hazard ratio (H.R.) of 1.24 (95% confidence interval [CI]: 1.03–1.49) for the group with the highest LAN exposure compared to the group with the lowest LAN exposure [87].

These findings suggest that sleep disruption and exposure to LAN may contribute to an increased risk of pancreatic cancer, highlighting the importance of maintaining healthy sleep patterns and minimizing circadian disturbances.

### 1.5. Avoidance of Toxins

**(A)** 
**Cigarette smoking:**


Cigarette smoking is a significant risk factor for pancreatitis and pancreatic cancer. It increases the likelihood of recurrent attacks of acute pancreatitis (A.P.) and contributes to the progression of A.P. to chronic pancreatitis (C.P.). The primary metabolites found in cigarette smoke, nicotine and NNK (N′-nitrosonornicotine and 4-(methylnitrosamino)-1-(3-pyridyl)-1-butanone), have detrimental effects on the acinar cells and zymogen secretion in the pancreas. These effects lead to functional and histological changes characteristic of acute pancreatitis. Moreover, smoking affects the pancreatic microvasculature through the nitric oxide pathway and can cause dysfunction in the CFTR (cystic fibrosis transmembrane conductance regulator), which influences ductal secretion [88].

It is worth noting that cigarette smoking often goes hand in hand with alcohol consumption, further amplifying the risk of developing pancreatitis.

When it comes to pancreatic cancer, a meta-analysis examining the impact of smoking on the risk of this disease revealed that current smokers have an odds ratio of 1.74 (with a 95% confidence interval of 1.61–1.87) compared to never-smokers [89]. This risk is particularly elevated in individuals who smoke a more significant number of cigarettes per day, with those smoking over 35 cigarettes per day having an odds ratio of 3.0 (95% CI 2.2–4.1) compared to never-smokers. However, the good news is that quitting smoking can substantially decrease this risk. Former smokers still face a slightly higher odds ratio of 1.2 (95% CI 1.11–1.29) for pancreatic cancer than never-smokers. However, what is interesting is that the risk of pancreatic cancer in former smokers diminishes over time after quitting. After 15–20 years, the risk returns to that of individuals who have never smoked [90,91]. This suggests that the negative impact of smoking on pancreatic cancer risk can be significantly mitigated in the long term after quitting smoking. Moreover, in areas where smoking prevalence has decreased, there has been a corresponding decrease in the proportion of pancreatic cancers attributed to smoking.

These findings underscore the strong association between cigarette smoking and the increased risk of pancreatitis and pancreatic cancer. Quitting smoking is of utmost importance in reducing the risk of developing these conditions, and over time, the risk of ex-smokers can approach that of individuals who have never smoked.

**(B)** 
**Alcohol:**


Alcohol abuse is commonly associated with the development of both acute and chronic pancreatitis. The risk of pancreatitis increases as alcohol consumption increases, and it has been found that consuming approximately five drinks per day (equivalent to 60 g of ethanol) raises the risk of developing pancreatitis [92,93]. However, it is essential to note that alcohol abuse alone does not always cause pancreatitis, as most heavy drinkers do not develop the condition. This suggests that additional factors or insults are involved in developing pancreatitis. Some suggested contributing factors include smoking, a high-fat diet, obesity, genetics, and infectious agents [92,94,95].

Alcohol use is strongly linked to the development of pancreatic cancer as well. There is a clear association between the development of pancreatic cancer and heavy alcohol intake, defined as consuming more than three drinks per day or 40 g of alcohol per day in dose/risk analyses. This level of alcohol consumption is associated with a 20% increase in pancreatic cancer risk. However, in large meta-analyses, no significant association has been observed among non-drinkers or occasional drinkers (consuming less than three drinks per day) [96]. A study from the European Prospective Investigation into Cancer and Nutrition (EPIC) found a positive association between alcohol intake and pancreatic cancer risk, particularly among men and heavy drinkers (consuming more than 60 g of alcohol per day) [97]. Furthermore, their study indicated that the risk was higher with beer and liquor consumption than with wine.

The potential mechanism behind the development of pancreatic cancer due to alcohol involves the metabolite of ethanol called acetaldehyde, which is released into the bloodstream. Acetaldehyde binds to DNA repair proteins, causing DNA damage and promoting the development of tumors [98]. Additionally, other ethanol metabolites, such as fatty acid ethyl esters, increase intracellular calcium release, which triggers toxicity in pancreatic acinar cells and initiates the process of pancreatic autodigestion by activating trypsinogen prematurely [99,100]. Recurrent injuries to pancreatic acinar cells can eventually lead to neoplastic transformation.

In summary, alcohol abuse is closely associated with the development of both pancreatitis and pancreatic cancer. These risks increase with higher levels of alcohol consumption. Additional factors such as smoking, diet, obesity, genetics, and infections may also contribute to the development of pancreatitis. In the case of pancreatic cancer, acetaldehyde and other ethanol metabolites play a role in promoting DNA damage and triggering toxic effects on pancreatic cells, ultimately leading to the development of cancer.

**(C)** 
**Refined sugar and sodas:**


Fructose derived from beverages is metabolized more rapidly than fructose in solid foods. In recent decades, there has been a significant increase in our consumption of added sugars, particularly in beverages containing corn-derived high-fructose syrup [101]. This trend has raised concerns regarding its potential impact on health, including the risk of pancreatic cancer. A study by Larson et al. revealed a greater risk of pancreatic cancer among heavy consumers of sugary soft drinks (more than two per day) and sweetened fruit juices compared to those who consumed such beverages sporadically [102]. Another meta-analysis conducted in 2012 demonstrated a positive association between a fructose intake of 25 g per day and a higher risk of pancreatic cancer. This association was held regardless of glycemic index, sucrose consumption, and high carbohydrate intake [103].

Fructose and glucose are metabolized differently in the body. While glucose utilizes a sodium-dependent transporter for absorption, fructose is absorbed through glucose transporter type 5 (GLUT5) in the small intestine and is primarily metabolized in the liver. When blood glucose levels are high, pancreatic beta-cells produce insulin, and glucose is stored as glycogen with the assistance of glucose transporter type 4 (GLUT4). However, GLUT5 is not regulated by insulin, and fructose uptake remains uncontrolled. This metabolic pathway promotes pyruvate decarboxylation, leading to acetyl-CoA synthesis, which contributes to lipogenesis. Moreover, fructose interferes with the insulin signaling pathway, resulting in insulin resistance. Obesity and diabetes mellitus, both significant risk factors for pancreatic cancer, can arise from this disruption [104].

Additionally, research has shown that pancreatic cancer cells prefer fructose over glucose in the non-oxidative pentose phosphate pathway (PPP). This pathway facilitates the production of five-carbon pentoses from six-carbon glucose, providing new substrates for RNA synthesis. The increased fructose utilization in nucleic acid synthesis leads to heightened uric acid production, affecting purine metabolism. In a study conducted by Hsieh et al., high levels of fructose were found to promote aggressive cancer development in mice with specific KRAS mutations, resulting in a higher grade of pancreatic lesions, increased expression of GLUT5, and more significant metastatic potential. In vitro experiments substituting glucose with fructose also led to the selective outgrowth of an invasive and drug-resistant subpopulation of cancer cells, intensifying cancer cell metastasis [105].

These findings suggest a potential link between high fructose consumption, particularly in the form of beverages, and an increased risk of pancreatic cancer. The metabolic differences between fructose and glucose and the preferential utilization of fructose by pancreatic cancer cells contribute to the detrimental effects observed. However, further research is needed to fully understand the complex relationship between fructose, pancreatic cancer development, and its progression.

**(D)** 
**Food preservatives:**


Ultra-processed foods are industrial products that consist mainly or entirely of substances derived from foods and additives. These foods typically contain minimal or no intact whole foods and are characterized by affordability, hyperpalatability, convenience, and high energy density [106]. The consumption of ultra-processed foods has been associated with various health risks, including an increased risk of pancreatic cancer.

One of the reasons ultra-processed foods are concerning is that they often contain carcinogenic components that result from food processing or packaging. For example, the processing of certain foods can lead to the formation of carcinogenic substances such as heterocyclic amines and acrylamide. Packaging materials, such as bisphenol, can also introduce potentially harmful substances. These carcinogenic components contribute to the overall risk associated with consuming ultra-processed foods [107,108,109].

Furthermore, the consumption of ultra-processed foods has been linked to an increased risk of type II diabetes mellitus and obesity, which are well-established risk factors for pancreatic cancer [110,111]. Studies have shown that these foods can contribute to weight gain and promote conditions that increase the likelihood of developing diabetes and obesity, ultimately raising the risk of pancreatic cancer.

A large prospective trial conducted by Zhong et al. involving 98,265 American adults demonstrated a positive association between ultra-processed food consumption and the risk of pancreatic cancer. The relationship between consumption and risk was found to be nonlinear in a dose–response manner [112].

Another randomized controlled trial conducted in an inpatient setting showed that increased energy intake, including carbohydrates and fats derived from ultra-processed foods, contributes to obesity, which can lead to an elevated risk of pancreatic cancer. Additionally, consuming ultra-processed foods displaces the intake of fresh fruits and vegetables, which are rich sources of dietary fiber that can help reduce the risk of cancer [113].

It is also worth noting that ultra-processed foods often contain cosmetic additives and preservatives, some of which have potential carcinogenic effects. For instance, sodium nitrate, a preservative commonly used in processed meats, can form nitrosamine in the stomach, a potent carcinogen associated with an increased risk of pancreatic cancer [114].

In light of these findings, it is recommended to prioritize limiting ultra-processed food consumption in order to decrease the prevalence of lifestyle diseases, obesity, and the incidence of fatal cancers like pancreatic cancer.

### 1.6. Social connection and volunteering

Social connection and volunteering can play significant roles in the lifestyle management of pancreatitis and pancreatic cancer. Social networks and social support were perceived as important factors that influence how everyday life is experienced during recovery and how to cope with disease symptoms [115].

Dealing with pancreatitis or pancreatic cancer can be emotionally challenging. Having a social support network can provide emotional support and reduce stress, anxiety, and depression associated with the condition. Being part of a supportive community can instill a sense of belonging and purpose, which is particularly important for individuals facing serious health challenges. Feeling connected to others and contributing to a cause can enhance overall well-being and quality of life [116,117].

Volunteering and social connections can also facilitate sharing information and experiences among individuals affected by panceratitis or pancreatic cancer. Connecting with others going through similar experiences, whether in-person or online, offers a platform for sharing experiences, coping strategies, and practical tips for managing the challenges associated with pancreatitis or pancreatic cancer. This knowledge exchange can empower patients with valuable insights into managing their condition effectively. It can offer practical assistance to patients, such as helping with transportation to medical appointments, running errands, or providing meals. These forms of support can alleviate some of the burdens associated with managing the illness [118]. Engaging in social activities and volunteering can motivate individuals to adopt healthier lifestyle choices, such as maintaining a balanced diet, exercising regularly, and avoiding harmful habits like smoking and excessive alcohol consumption. These lifestyle modifications can positively impact the management of pancreatitis and pancreatic cancer.

Overall, social connection and volunteering can complement medical treatment and lifestyle management strategies for pancreatitis and pancreatic cancer by providing emotional support, practical assistance, and opportunities for advocacy and awareness-raising. They are crucial in promoting holistic well-being and improving the quality of life for individuals affected by these conditions.

### 1.7. Additional Factors Influencing the Occurrence of Pancreatitis and Pancreatic Cancer

**(A)** 
**Microbiota**


The human body harbors many complex communities of microorganisms known as the microbiota, which play a crucial role in various aspects of health. These microbiota contribute to nutritional and hormonal balance, regulate inflammation, detoxify compounds, and produce metabolites that can have metabolic effects [119]. However, emerging research indicates that microbiota can also influence carcinogenesis and treatment outcomes. They can promote inflammatory responses, alter the tumor-immune microenvironment, affect tumor metabolism, and modulate tumor sensitivity to drugs.

The oral microbiota, which includes bacteria that can cause periodontal disease and tooth loss, has been implicated in pancreatic carcinogenesis. Oral microbes can translocate or disseminate to the pancreas, leading to inflammation [120,121,122]. Among the oral bacteria associated with pancreatic ductal adenocarcinoma (PDAC), *Porphyromonas gingivalis* (*P. gingivalis*) is considered a key pathogen. *P. gingivalis* secretes enzymes called peptidyl-arginine deiminases (PADs) that can degrade arginine, potentially resulting in p53 and K-ras gene mutations [123].

The gut microbiota, a complex ecosystem, breaks down pancreatic enzymes secreted into the intestine. While the antibacterial activity of pancreatic juice helps protect the pancreas from retrograde infections, gut microbes can reach the pancreas through the circulatory system or the biliary/pancreatic duct, leading to inflammation and carcinogenesis [124,125].

Several studies have suggested *Helicobacter pylori* (*H. pylori*), a bacterium associated with gastric ulcers and stomach cancer, as a possible risk factor for PDAC [126,127,128]. However, conflicting results have been reported, with some studies finding no relationship or inverse association between *H. pylori* and PDAC [129,130].

Hepatitis B virus (HBV) and hepatitis C virus (HCV), which primarily infect the liver and are linked to hepatocellular carcinoma (HCC), can also be detected in extrahepatic tissues, including the pancreas [131]. HBsAg and HBcAg have been detected in the cytoplasm of pancreatic acinar cells, and elevated serum and urine pancreatic enzymes have been found in individuals with chronic HBV infection [132]. HBsAg has also been associated with the development of pancreatitis among patients with HBV infection, which suggests that HBV-related pancreatitis might be a precursor of PDAC [133].

Bile is a sterile hepatobiliary solution, and the most common microbes infecting bile are *Enterobacter* and *Enterococcus* spp. Using bacterial culture and genetic sequence analysis, a study by Markawa et al. showed increased levels of antibodies against *Enterococcus faecalis* capsular polysaccharide (CPS) in the serum of PDAC and chronic pancreatitis patients compared with healthy subjects, which may indicate that infection with *E. faecalis* is involved in the progression of pancreatitis-associated PDAC [134]. A cross-sectional study by Serra et al. showed an increased risk of pancreatic head adenocarcinoma with biliary infection with *Pseudomonas* spp. and *Escherichia coli* [135].

Traditionally, the pancreas was considered a sterile organ, but recent research has identified an increased presence of bacteria in pancreatic ductal adenocarcinoma tissues compared to normal pancreas tissue. Certain bacteria, such as *Gammaproteobacteria*, have been detected in the pancreas of patients with gemcitabine-resistant advanced pancreatic cancer, suggesting a potential role in modulating tumor sensitivity to chemotherapy [136].

Various factors, including diabetes mellitus, dietary intake, smoking, and obesity, can influence microbiota composition. These factors can impair the immune system, induce inflammation, and lead to microbial dysbiosis, which can impact the development and progression of PDAC [137,138]. Modulating the microbiota may hold promise for enhancing drug efficacy and reducing toxicity. For example, a low-fat, high-fiber diet has been associated with increased microbiome diversity and an abundance of beneficial nutrients [139]. Future studies should explore microbiome-based biomarkers and investigate the impact of dietary and lifestyle changes on treatment responses and patient outcomes in PDAC.

**(B)** 
**Non-alcoholic fatty pancreatic disease (NAFPD)**


Obesity, particularly abdominal obesity, is associated with insulin resistance, which can lead to pancreatic steatosis and non-alcoholic fatty pancreatic disease (NAFPD). Previous studies have suggested that a high-fat diet can cause fatty infiltration of the pancreas, potentially leading to a loss of β-cell mass and function and increasing the risk of developing diabetes. Adipose tissue is not simply a storage depot for triglycerides (T.G.s), but it actively releases various factors, such as cytokines, metalloproteinases, and adipokines, which can induce inflammation and insulin resistance [140].

Researchers Smits and Van Geenen have proposed that increased pancreatic fat associated with obesity contributes to a higher incidence and more severe episodes of acute pancreatitis due to an imbalance of adipocytokines, leading to inflammation [141]. A prospective study by Fuji et al. found that fat accumulation in the pancreas was a significant risk factor for chronic pancreatitis [142]. Furthermore, studies have observed fatty degeneration and fibrosis in the pancreatic tissue surrounding most cases of pancreatic adenocarcinoma, indirectly suggesting that NAFPD might eventually lead to pancreatic cancer, similar to how non-alcoholic fatty liver disease (NAFLD) can cause liver cancer [143]. Several studies on the association between non-alcoholic fatty pancreas disease (NAFPD) and metabolic syndrome have also demonstrated that ectopic fat can contribute to pancreatic cancer and insulin resistance [144,145,146].

Lifestyle modifications, including weight reduction, exercise, and dietary changes, can improve pancreatic fatty infiltration associated with metabolic syndrome. In a study by Honka et al. [146], morbidly obese patients who underwent bariatric surgery showed that insulin resistance related to fatty pancreas was reduced after weight loss. There was a notable decrease in pancreatic fat volume (*p* < 0.01), fatty acid uptake (*p* < 0.05), and blood flow (*p* < 0.05) post bariatric surgery [147]. Diet and exercise are powerful tools in improving ectopic fat deposition and the organ’s function in which the ectopic fat is deposited. Lifestyle modifications such as reducing caloric intake and meat consumption may benefit patients with NAFPD [148]. Vegetarian and vegan diets are effective for weight loss, particularly for reducing visceral fat and subfascial fat in muscle tissue involved in glucose homeostasis [148,149]. Therefore, diet and lifestyle interventions deserve attention as preventive measures for obesity and type II diabetes mellitus, reducing the risk of recurrent idiopathic pancreatitis and pancreatic cancer.

**(C)** 
**Gallbladder disease**


Gallstones are the leading cause of acute pancreatitis (A.P.). The prevalence of gallstones in the U.S. adult population is approximately 7% [150]. The main contributing factors to cholelithiasis (gallstone formation) are cholesterol supersaturation, crystallization, and gallbladder dysmotility, often occurring together [151].

Research has indicated that the risk of gallstones is positively associated with the intake of meat, energy, fat, and saturated fat. In contrast, it is negatively related to the intake of vegetables and fiber in both Western and Asian populations [152,153,154]. The consumption of red meat inhibits bile acid transporters, promotes cholesterol gallstone formation, and increases the risk of gallstone disease in individuals with a high meat intake [152,155]. Additionally, meat is a significant source of total fat and saturated fat, which can decrease insulin sensitivity and contribute to gallbladder disease and dysmotility, which are linked to an increased risk of gallstones [156]. In individuals with insulin resistance and hyperinsulinemia, the activity of HMG-CoA reductase, the rate-limiting enzyme in hepatic cholesterol synthesis, is elevated. This leads to an increased cholesterol saturation index in the bile and subsequent cholesterol gallstone formation [157,158].

Previous studies have demonstrated a negative association between vegetable and fiber intake and the risk of gallstone formation in Western populations [159,160,161,162]. This suggests that a diet rich in vegetable fiber may protect against gallstone disease. Insoluble fiber intake has been shown to decrease intestinal transit time, reduce biliary deoxycholic acid levels, and decrease the cholesterol saturation index [163].

Vitamin C acts as a cofactor for the enzyme 7-hydroxylase, which converts cholesterol into bile acids. By doing so, it decreases the pathogenicity (stone-forming potential) of bile. In animal studies [164,165], vitamin C supplementation has been found to inhibit cholelithiasis and accelerate the conversion of cholesterol to bile salts [165]. In a study involving patients with gallstones, daily supplementation with 2 g of vitamin C for two weeks decreased the pathogenicity of bile [166].

Therefore, increasing the consumption of fruits and vegetables as part of a healthy diet can enhance dietary fiber intake, reduce intestinal transit time, and subsequently decrease cholesterol supersaturation and the formation of gallstones and gallstone-related pancreatitis.

**(D)** 
**Hydration and pancreatitis**


Intravenous fluid resuscitation is an essential part of initial supportive therapy in acute pancreatitis (A.P.) because untreated pancreatic hypoperfusion can lead to poor outcomes such as pancreatic necrosis and death [167]. The microcirculatory disturbances in A.P. differ from those in simple hypovolemia caused by trauma or bleeding. In A.P., these disturbances are caused by the systemic inflammatory response syndrome (SIRS), characterized by the overexpression of inflammatory mediators. These mediators can injure the endothelium, increase capillary permeability, and result in fluid sequestration and capillary leak syndrome. Adequate fluid resuscitation in severe A.P. aims to replenish blood volume, stabilize capillary permeability, modulate the inflammatory response, and maintain intestinal barrier function [168].

Post-endoscopic retrograde cholangiopancreatography pancreatitis (PEP) is the most common serious adverse event following this procedure, with an incidence ranging from 4% to 10% and a mortality rate reaching 0.7% [169]. Pathophysiological mechanisms contribute to PEP, including pancreatic microvasculature hypoperfusion, ischemic injury, and zymogen activation from acidemia. Periprocedural hydration with lactated Ringer’s solution has been associated with a lower incidence of PEP. In a randomized, controlled, double-blind clinical trial involving 150 patients, Shaygan-Nejad et al. found that aggressive hydration was associated with a lower incidence of PEP compared to standard hydration in patients undergoing ERCP without the prophylactic administration of rectal nonsteroidal anti-inflammatory drugs (NSAIDs) (5.5% versus 22.7%, *p*-value = 0.002) [170,171].

During prolonged endurance activities like marathons, individuals can lose significant amounts of fluid through sweating. If not adequately replenished, this fluid loss can result in dehydration and pancreatitis in susceptible individuals [172,173].

Therefore, increasing hydration and maintaining sufficient fluid intake to keep urine clear or light yellow may help prevent the recurrence of or minimize the intensity/frequency of idiopathic pancreatitis in high-risk patients.

## 2. Discussion

Assessing the various risk factors associated with pancreatitis and pancreatic cancer can be challenging due to the complexity of these conditions. Many risk factors often work together to increase an individual’s predisposition to developing pancreatitis and pancreatic cancer, and there is a lack of studies examining these factors’ combined effects. While pancreatic cancer is less common than other types of cancer, prevention through lifestyle changes is crucial due to its aggressive and deadly nature and the difficulties in diagnosis.

Based on current data, adopting a comprehensive approach using the principles of lifestyle medicine can be beneficial in reducing the incidence of pancreatitis and pancreatic cancer. These principles include the following:Consuming a low-fat, whole-food, plant-predominant diet: This involves avoiding processed foods, animal products, and saturated fats and focusing on consuming various fruits, vegetables, beans, nuts, seeds, and whole grains. Such a diet provides essential nutrients and phytochemicals that support overall health and reduce the risk of chronic diseases, including pancreatic conditions.Engaging in regular physical activity: Physical activity is crucial in maintaining a healthy weight, improving insulin sensitivity, and reducing inflammation. Regular exercise has been associated with a lower risk of various cancers, including pancreatic cancer.Avoiding toxins: It is crucial to steer clear of tobacco and alcohol consumption, as they are known risk factors for pancreatic diseases. Tobacco use, in particular, has a strong association with pancreatic cancer.Ensuring adequate sleep: Sufficient sleep is essential for overall well-being and maintaining a healthy immune system. Poor sleep patterns and chronic sleep deprivation have been linked to increased inflammation and a higher risk of various diseases.Managing stress: Chronic stress can harm health, including increased inflammation. Stress management techniques, such as meditation, yoga, or engaging in hobbies, can help reduce stress levels and promote overall well-being.Engaging in social support networks and volunteering enhances quality of life and promotes overall well-being.Maintaining an excellent microbial flora: Through a balanced and diverse fiber-rich diet, limiting unnecessary antibiotic use, and managing stress, an excellent microbial flora can be maintained. This supports overall gut health, which may also contribute to pancreatic health.Staying well-hydrated: Maintaining good hydration can indirectly reduce the risk of pancreatic disease in susceptible individuals.

By adopting these lifestyle measures, individuals can reduce their exposure to triggers that stimulate inflammation, which can contribute to the development of pancreatitis and pancreatic adenocarcinoma. Moreover, such lifestyle changes can improve the overall quality of life while limiting the risk of chronic diseases. Further research focusing on these aspects of lifestyle medicine and their impact on preventing pancreatitis and pancreatic cancer in order to better understand their effectiveness and develop more targeted interventions is highly recommended.

## Figures and Tables

**Figure 1 diagnostics-14-00614-f001:**
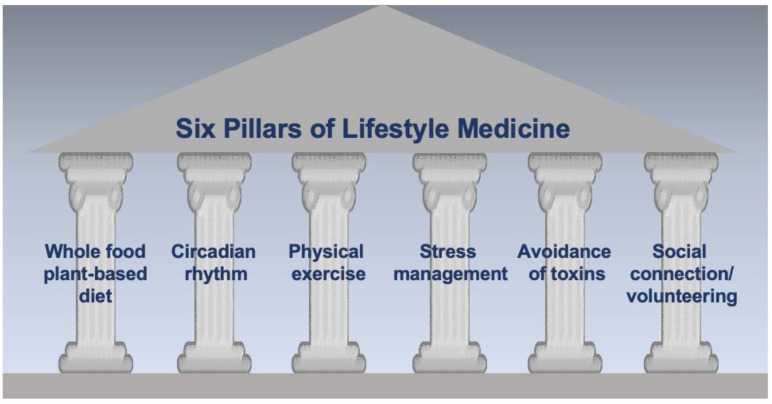
The six pillars of lifestyle medicine.

**Table 1 diagnostics-14-00614-t001:** The six pillars of lifestyle medicine and how they affect the prevention and treatment of pancreatitis and pancreatic cancer.

Six Pillars of Lifestyle Medicine	Components of Each Pillar	Impact on the Body
Whole food plant- based diet	1. Phytates	Anticancer and immunomodulatory properties
	2. Methionine restriction	Anticancer effect: cancer cell require methionine for growth and metabolism
	3. Plant protein vs Animal protein	Plant protein—Anticancer effect by lowering insulin like growth factor 1
	4. Fiber	Anticancer effect: Alters intestinal microbiota which regulate immune and inflammatory responses.
	5. Curcumin	Antioxidant, anti-inflammatory, and anticancer activities.
	6. No meat	Reduce intake of a source of carcinogenic heterocyclic amine (HCA) and polycyclic aromatic hydrocarbons (PAH).
	7. No animal fat	Reduce intake of saturated fat, which can lead to insulin resistance associated with elevated risk of pancreatic cancer
	8. No poultry	Potential for cancer from the poultry viruses
	9. No heme ron	Heme iron leads to formation of reactive oxygen species which promotes carcinogenesis.
Physical activity	Anticancer effect by reducing abdominal fat deposits inducing metabolic improvements in glucose tolerance and insulin sensitivity.	
Stress management, mindfulness/meditation	Stress leads to overexpression of β-adrenergic signaling which promotes carcinogenesis	
Restorative sleep to maintain circadian rhythm	Disruption leads to oncogenesis by affecting the immune, cellular, metabolic, and endocrine functioning.	
Avoidance of toxins	1. Cigarette smoking	Metabolites causes pancreatitis and increase the risk of pancreatic cancer
	2. Alcohol	Metabolite acetaldehyde causes DNA damage and promote carcinogenesis
	3. Refined sugar intake and sodas	Causes insulin resistance—a risk factor for pancreatic cancer. It provides fructose, which is a significant substrate for tumorigenesis.
	4. Food preservatives, coloring agents, bleach, etc.	Contains carcinogenic components, cause inflammation, adversely affect gut microbiome
Social connection and volunteering	1. Emotional support	Reduce stress, anxiety and depression
	2. Information sharing and education	Empower patients with valuable insights into managing pancreatitis and pancreatic cancer
	3. Peer support	Provide coping strategies and practical tips in the management
	4. Sense of belonging and purpose	Enhance overall well-being and quality of life
	5. Encouragement for Healthy Lifestyle choices	Balanced diet, regular exercise and avoiding harmful habits like smoking and alcohol consumption

## Data Availability

No new data was created.

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
