# Peer review of "Lifestyle Medicine for the Prevention and Treatment of Pancreatitis and Pancreatic Cancer"

_diagnostics, 2024, doi:10.3390/diagnostics14060614_

Round 1

Reviewer 1 Report

Comments and Suggestions for Authors

Dear authors,

I would like to congratulate for this manuscript; it is outstanding thanks to the completeness of the treatment of the topic and the order apllied. I think it deserves publication because it constitutes a very useful document for all physicians, even and above all non-experts in the field of pancreatitis, who look in high level journals valid documents for their updating on such a socially relevant topic.

Author Response

Thank you for your response. Appreciate it!

Reviewer 2 Report

Comments and Suggestions for Authors

The work is well organized, has an abstract that is clear and synthetic. This is a narrative review, succinct and extensively referenced (167 references)

The chapters are as expected;

Introduction – a summary review of acute and chronic pancreatitis and adenocarcinoma with identification of risk factors and potential protective aspects. It also addresses “Lifestyle Medicine”

Table 1 summarizes the 6 pillars of “lifestyle medicine” and the components that impact the body, which are then discussed in corresponding chapters (2-7). The chapters are objective and synthetic. In chapter 8, a discussion of additional factors follows; Non-alcoholic fatty pancreatic disease, Gall bladder disease and Hydration, perhaps the latter in a subchapter with a first paragraph a little out of phase with the rest, mixing the issue of the shock of severe pancreatitis with preventive hydration of pancreatitis post endoscopic retrograde cholangiopancreatography. The final discussion is detailed and summarizes the therapeutic principles arising from the analysis, although it omits the issue of hydration.

My only suggestion is a reflexion about the oportunity of the first paragraph of the sub-chapter 8(c) Hydration anda Pancreatitis

Author Response

Thank you for your expert response. I have added more details into the hydration and pancreatitis topic and discussion.